# The Rhizosphere Functional Microbial Community: A Key Driver of Phosphorus Utilization Efficiency in Karst Forest Plants

Chunjie Zhou [1], Danmei Chen [1,2,*], Lipeng Zang [1,2], Guangqi Zhang [1,2], Qingfu Liu [1,2], Mingzhen Sui [1,2], Yuejun He [1], Shasha Wang [1], Yu Dai [1], Lidong Wang [3], Ruxia Bai [3], Ziyun Feng [3] and Fachun Xiang [3]

[1] College of Forestry, Guizhou University, Guiyang 550025, China; zz15286269622@163.com (C.Z.); cafzanglp@163.com (L.Z.); gqzhang1@gzu.edu.cn (G.Z.); qingfuliu@gzu.edu.cn (Q.L.); cafsmz@163.com (M.S.); hyj1358@163.com (Y.H.); lishawang0419@163.com (S.W.); daiyu_0207@163.com (Y.D.)

[2] Guizhou Libo Karst Forest Ecosystem National Observation and Research Station, Libo 558400, China

[3] Mianyang Agricultural Products Quality Safety Inspection and Testing Center, Mianyang 621000, China; wanglidong000@163.com (L.W.); 18306038510@163.com (R.B.); 15008218082@163.com (Z.F.); xfc_0417@163.com (F.X.)

\* Correspondence: dorischan0808@163.com; Tel.: +86-18608149225

**Abstract:** Microorganisms play a pivotal role in transforming and making phosphorus (P) available in soil through various mechanisms. However, their specific contributions to alleviating P limitation and enhancing P utilization efficiency in plants within the context of a P-deficient karst ecosystem remains unclear. In this study, eco-stoichiometric methods were employed to evaluate the P utilization efficiency of plants grown in the surveyed karst forest located in Guizhou Province, China. Metagenomic sequencing was utilized to further explore the functional genes and microorganisms involved in soil P cycling. The N:P ratio for 18 out of the 20 surveyed plants exceeded 16, indicating widespread P limitation in karst plants. Among them, plants with high P utilization efficiencies (*Nandina domestica* Thunb.; *Mahonia bodinieri* Gagnep.; *Pyracantha fortuneana* (Maxim.) Li) exhibited higher relative abundances of genes involved in soil P cycling compared to plants with low P utilization efficiencies (*Tirpitzia sinensis* (Hemsl.) Hallier f.; *Albizia kalkora* (Roxb.) Prain; *Morella rubra* Lour.), indicating greater potentials within their rhizosphere microbiomes for soil P transformation. The relative abundance of these functional genes had a significant and positive effect on plant P utilization efficiencies. Structural equation modeling further indicated that microbial P cycling gene abundance directly drove the increase in plant P utilization efficiencies. Specifically, genes involved in soil organic P mineralization (*G6PD*, *suhB*, *phoD*, *ppx*) and the P uptake and transform system (*pstS*, *pstA*, *pstB*, *pstC*) contributed to the enhancement of plant P utilization efficiencies. Soil microbial communities involved in P cycling were predominately attributed to Proteobacteria (45.16%–60.02%), Actinobacteria (9.45%–25.23%), and Acidobacteria (5.90%–9.85%), although their contributions varied among different plants. The rhizosphere functional microbial community can thus alleviate P limitation in karst plants, thereby enhancing plant P utilization efficiencies. This study investigated the strong synergism between karst plants and rhizosphere microorganisms and their associated underlying mechanisms from genetic and microbial perspectives.

**Keywords:** eco-stoichiometry; metagenomics; functional genes; soil phosphorus cycling





## 1. Introduction

Phosphorus (P) is a crucial macronutrient that is widely distributed and plays an indispensable role in numerous cellular processes, including cell metabolism, molecular building, energy storage and transfer, signal transmission, and photosynthetic regulation [1–3]. Based on data from 41 publications covering 258 different soils, it has been found that the reserves of P in soils are abundant enough to support global maximum agricultural production for approximately 350 years [4]. While there is an abundance of P in soils, it primarily exists in the forms of inorganic and organic forms that are unavailable to plants and microorganisms.

Therefore, despite its importance, P is often a limiting factor in various terrestrial ecosystems, and its scarcity significantly affects ecological processes and the overall productivity of these ecosystems [5–7].

Most of the P in soil is sequestered in recalcitrant minerals and organic compounds, with only a small amount being directly utilized by microorganisms and plants in the form of inorganic orthophosphate [8]. Microorganisms play a crucial role in transforming and making P available in the soil through processes such as mineralization, solubilization, and symbiotic associations. Mineralization involves the production of phosphatase enzymes by microorganisms, which break down organic P compounds into inorganic forms, primarily phosphate (Pi), thereby making P accessible for plant uptake [9]. Solubilization occurs when certain microorganisms possess the capability to dissolve insoluble forms of P in the soil. They release organic acids and other compounds that aid in the dissolution of P-bound minerals, thereby enhancing their availability for plant uptake [10]. Symbiotic associations are observed in certain microorganisms, such as mycorrhizal fungi, which form mutually beneficial connections with plant roots. They do so by secreting enzymes that aid in solubilizing and mobilizing P from organic and mineral forms [11]. As a result, microorganisms mediate soil P cycling and provide a significant proportion of soil-available P. Specifically, the microbial utilization of unavailable P is primarily facilitated by two functional gene groups: organic P mineralization and inorganic P solubilization [12]. In cases where bioavailable P is deficient, microorganisms increase the expressions of genes responsible for producing organic anions that aid in the solubilization of inorganic P, or of enzymes that facilitate the mineralization of organic P. Organic anions play a crucial role through processes such as acidification, chelation, and exchange reactions, ultimately leading to the release of sparingly available inorganic P forms [13–15]. Additionally, microorganisms facilitate the conversion of soil organic P compounds into bioaccessible forms through the action of a range of organophosphorus hydrolases, including phosphomonoesterase, phosphodiesterase, phosphonatase, and phytase [10]. Furthermore, variations in soil P availability trigger the expression of microbial functional genes involved in the uptake and transport of P, as well as the regulation of the P-starvation response system [12,16–18].

Previous studies have demonstrated that most soil microbes are primarily limited by carbon (C) [19] and are readily attracted to plant exudates near the roots, leading to their rapid proliferation and enrichment [20]. As a result, the soil in this area becomes distinct compared to the surrounding bulk soil, with the presence of root exudates enhancing microbial biomass and activity [21,22]. In essence, significant disparities, encompassing microbial function, diversity, and interactions with plants, manifest in the rhizosphere as opposed to other soil microorganisms [22]. The rhizosphere region plays a crucial role in plant–microorganism interactions, making significant contributions to plant nutrient acquisition and growth [23,24]. In addition to the direct impact of the quantity and composition of plant root exudates [25], two other important factors that determine the structure and function of the microbial community present in the vicinity of plant roots are plant species and soil type [26–29]. Different plant species produce varying amounts and qualities of C resources, shaping the composition of the microbial community in the rhizosphere [30–32]. Consequently, distinct plant-specific microbial communities can emerge, even under similar growth conditions [33–35]. However, prior research has highlighted that the interactions between plant species and soil microbes may be contingent on specific contexts [36], and in certain instances, such relationships may not manifest [37,38]. For instance, Bezemer et al. [36] noted significant disparities in microbial communities within different monocultures in chalk soil, whereas no discernible effect of plant species was observed in sandy soils. Similarly, Kielak et al. [38] identified only modest influences of plant species on rhizosphere communities, and Singh et al. [37] found no conclusive evidence of plant species exerting a discernible influence on soil microbial communities in the rhizospheres of various grass species. Soil type, in conjunction with plant species, also exerts a substantial influence on the structure and function of rhizosphere-associated microbial communities. Different soil types are known to harbor distinct microbial popula-

tions. For instance, Landesman et al. [39] analyzed 700 soil samples from multiple forests in the eastern United States, revealing that variations in soil properties, particularly pH, associated with different tree species, led to discernible differences in bacterial communities across various sites. Moreover, in the context of secondary succession in subtropical forests, the soil's C:N ratio directly impacted bacterial community composition and diversity, while pH significantly influenced fungal community composition and diversity [40]. This underscores the nuanced nature of plant–microbe interactions, indicating that they are inherently context-dependent and cannot be evaluated solely based on a single aspect.

Karst ecosystems, accounting for approximately 15% of the Earth's surface, develop from carbonate rocks such as limestone or dolomite [41]. These ecosystems possess distinctive ecological and chemical properties, characterized by a shallow soil layer, a relatively high soil pH, and substantial levels of Ca and Mg, but low levels of bioaccessible P [42–44]. As a result, the soil microbial communities in karst regions may differ from those in conventional ecosystems. Previous research has indicated that rhizosphere microorganisms play a pivotal role in facilitating the adaptation of karst plants to these challenging soil environments [45,46]. For instance, Tang et al. [46] demonstrated that fungal communities in the rhizosphere contribute significantly to the adaptation of *Themeda japonica* to karst rocky desertification, ultimately enhancing plant growth and ecological performance in karst areas. Similarly, a study by Xiao et al. [45] revealed that in karst areas with elevated hydrothermal levels, arbuscular mycorrhizal fungi and diazotrophs enhance nutrient absorption and transport in plants during the vegetation recovery process. However, in karst ecosystems with a server shortage of soil-available P, studies investigating plant P utilization efficiency as influenced by rhizosphere P functional microbial communities remain scarce.

In karst soils, P is typically sequestered in poorly soluble primary minerals, such as carbonate rock, or bound to calcium ($Ca^{3+}$) and magnesium ($Mg^{2+}$) ions, resulting in limited P availability [47]. Consequently, the constrained plant growth caused by low levels of soil-available P is a widespread occurrence in karst regions, presenting a significant challenge to agricultural production and ecological restoration efforts in degraded karst ecosystems. Plant–microbe interactions exhibit a general tendency to enhance soil P availability and improve plant P uptake by modifying the abundance and diversity of microorganisms [48]. To acquire the available P in the soil, plants strategically allocate photosynthate and other nutrients to the rhizosphere, thus promoting the release of accessible P through the regulation of microbial communities and their activities [49,50]. Therefore, in karst areas, the functions of rhizosphere microbial P cycling become pivotal in meeting plant P demands and alleviating P deficiency in the soil, contributing to the enhanced P utilization efficiency of plants in these challenging karst environments. Nonetheless, it remains uncertain whether specific correlations exist between plant P utilization efficiency and the associated rhizosphere microbial communities, as well as their underlying mechanisms in karst soils. To address this knowledge gap, we conducted an experiment in a typical karst forest located in Maolan County, Guizhou Province, covering a survey area of 10,000 $m^2$. The leaves of young woody plants were collected to determine nutrient contents and conduct an evaluation via P:N stoichiometry, which serves as an indicator of the plant P utilization efficiency (PPEI) (a higher P:N ratio indicates a higher plant P utilization efficiency) [51]. Subsequently, rhizosphere soil samples were also collected to assess their physicochemical properties and the composition of soil functional genes and microorganisms involved in P cycling. Our hypotheses were as follows: (a) differences may exist in the rhizosphere microbial P cycling communities across plants with different P utilization efficiencies; (b) there would be significant correlations between rhizosphere microbial P cycling potentials and plant P utilization efficiencies; and (c) the taxonomic compositions of rhizosphere microorganisms involved in P cycling may differ across plants species.

## 2. Material and Methods

### 2.1. Study Site Description

The study site was located in Maolan County, Guizhou Province, and serves as a representative sample of a karst ecosystem (25°09′–25°20′ N, 107°52′–108°05′ E, 550–850 m above sea level). The study site experiences a humid subtropical monsoon climate, characterized by an average annual temperature of 15.3 °C, an annual relative humidity of 83%, and an annual precipitation of 1752 mm. Around 80% of the annual precipitation occurs between April and October. The soil in the study site was dominated by black lime soil, while the dominant vegetation consists of a mixed forest of subtropical evergreen and deciduous broad-leaved trees.

### 2.2. Plant Survey and Plant Nutrients Analysis

In May 2022, a survey plot with an area of 10,000 m$^2$ was chosen in the study site and then surveyed for plants following the guidelines proposed by Condit [52]. Given the substantial variations in soil properties across diverse spatial conditions and the number of root exudates released from plant roots at various stages of plant development, which profoundly influence the rhizosphere microbial community [53], in this study, various plants at identical growth stages were chosen, all of which were cultivated on soil surfaces with a relatively consistent thickness on downhill slopes. Specifically, plants with a diameter at breast height between 1.0 cm and 2.0 cm and with a height between 1.0 m and 1.5 m (mostly shrubs and saplings) were investigated and recorded. Specifically, 20 matured and fully expanded healthy leaves were collected from each plant for later nutrient analysis. The collected plant leaves were dried and ground for the nutrient analysis. Plant carbon (C) and nitrogen (N) contents were measured using an elemental analyzer (Elemental, Langenselbold, Hess, Germany); phosphorus (P) and potassium (K) contents were measured using the molybdate colorimetry method [54] and the flame photometer method [54]; and the calcium (Ca) content was measured using an atomic absorption spectrophotometer (ICE 3500, Thermo Scientific, Waltham, MA, USA) after $HNO_3$-$HClO_4$ digestion [55].

A total of 20 plants were selected and analyzed for nutrient contents, and the results are shown in Table S1. Plant C contents were the highest (39.16%–56.53%, average of 46.46%), followed by N (1.21%–3.72%, average of 1.91%), Ca (0.59%–2.16%, average of 1.25%), K (0.24%–0.57%, average of 0.35%), and P (0.04%–0.28%, average of 0.09%). We further calculated the leaf N:P ratio and the P:N ratio. The N:P ratio was used to reveal the limiting element for plant growth (N or P) [56,57]. While the P:N ratio represents the plant P utilization efficiency index (PPEI) growing in terrestrial ecosystems [51]. Therefore, we chose 3 plants with a high PPEI (*ND*, *Nandina domestica* Thunb.; *MB*, *Mahonia bodinieri* Gagnep.; and *PF*, *Pyracantha fortuneana* (Maxim.) Li) and 3 plants with a low PPEI (*TS*, *Tirpitzia sinensis* (Hemsl.) Hallier f.; *AK*, *Albizia kalkora* (Roxb.) Prain; and *MR*, *Morella rubra* Lour.) from the 20 investigated plants based on the results for subsequent analysis.

### 2.3. Soil Sampling and Analyses

Rhizosphere soil samples of 50 g were taken from each plant in August 2022 (a total of 20 soil samples) and air-dried for the determination of chemical properties. Specifically, soil pH was measured usinfg a glass meter; the total N (TN) and total P (TP) were measured using the Kjeldahl method [58] and the molybdate colorimetry method [54]; the total Ca (TCa) was measured using an atomic absorption spectrophotometer (ICE 3500, Thermo Scientifc, Waltham, MA, USA) after $HNO_3$-$HClO_4$ digestion [55]; soil organic C (SOC) was measured using the dichromate oxidation method [59]; and soil-available N (AN), soil-available P (AP), and soil-available Ca (ACa) were measured using the alkaline hydrolysis diffusion method [54], the molybdate colorimetry method [54], and the atomic absorption spectrometry method [55], respectively. In addition, fresh soil samples collected from the *ND*, *MB*, *PF*, *TS*, *AK*, and *MR* rhizospheres were frozen in liquid $N_2$ immediately for subsequent microbial analysis.

### 2.4. Soil DNA Extraction and Metagenomic Sequencing

Soil DNA extraction and subsequent metagenomic analysis were conducted as follows. (i) DNA extraction: 1.0 g of the collected soil samples was processed for DNA extraction using the FastDNA® Spin Kit for Soil (MP Biomedicals, Irvine, CA, USA). The extraction process involved the disruption of soil aggregates, cell lysis, and purification of the DNA. (ii) Quality assessment: the extracted DNA was assessed for quantity and quality using the TBS-380 micro-fluorometer (TurnerBioSystems, Sunnyvale, CA, USA) and the NanoDrop 2000 ultra micro-spectrophotometer (Thermo Scientific, Waltham, MA, USA), respectively. This step ensured that a sufficient amount of high-quality DNA was available for downstream analysis. (iii) Library preparation: the extracted DNA was then subjected to library preparation, which involved fragmenting the DNA (about 400 bp), adding adapters, and amplifying the DNA fragments via PCR (polymerase chain reaction). (iv) Sequencing: the prepared DNA libraries were sequenced using Illumina NovaSeq (Illumina Inc., San Diego, CA, USA) and this step generated millions of short DNA sequences, collectively representing the metagenome of the soil sample.

### 2.5. Quality Control and Genome Assembly

A total of 959,221,902 raw reads were initially obtained from the metagenome sequencing. These raw reads were processed to generate clean reads by removing adaptor sequences and trimming low-quality reads. The cleaning process involved eliminating reads with N bases, setting a minimum length threshold of 50 bp, and applying a minimum quality threshold of 20. The software tool FASTP 0.20.0 [60] was utilized for this purpose. After quality control, 933,426,482 clean reads were obtained and assembled to contigs using MEGAHIT [61] (https://github.com/voutcn/megahit, accessed on 4 October 2023). Contigs with a length of 300 bp or longer were chosen as the final assembly output.

### 2.6. Bioinformatic and Functional Analysis

The MetaGene tool (a C++ meta-program genertion tool) [62] (http://metagene.cb.k.u-tokyo.ac.jp/, accessed on 13 October 2023) was employed to detect open reading frames (ORFs) within the contigs. ORFs that were predicted and had a length of 100 bp or greater were extracted. These identified ORFs were then translated into amino acid sequences using the NCBI translation table (http://www.ncbi.nlm.nih.gov/Taxonomy/taxonomyhome.html/index.cgi?chapter=tgencodes#SG1, accessed on 13 October 2023). We developed a non-redundant gene catalog using CD-HIT [63] (http://www.bioinformatics.org/cd-hit/, accessed on 13 October 2023), applying a 90% sequence identity and 90% coverage threshold. After quality control, the reads were aligned to the non-redundant gene catalog using SOAPaligner [64] (http://soap.genomics.org.cn/, accessed on 13 October 2023), with a 95% identity cutoff. The gene abundance in each sample was assessed based on the alignment results. For annotation, representative sequences from the non-redundant gene catalog were compared against the NCBI NR database using BLASTP implemented in DIAMOND v0.9.19, with an e-value cutoff of $1 \times 10^{-5}$ [65] (http://www.diamondsearch.org/index.php, accessed on 23 October 2023). Additionally, taxonomic annotations were performed using Diamond against the KEGG database for Kyoto Encyclopedia of Genes and Genomes (KEGG) annotation (http://www.genome.jp/keeg/, accessed on 23 October 2023), also with an e-value cutoff of $1 \times 10^{-5}$. In the context of soil P cycling, we focused on the genome functional analysis of protein-coding genes. To this end, we referred to previous publications and selected 119 functional genes with their corresponding KO numbers from the KEGG database. These genes were attributed to four categories: (a) organic P mineralization; (b) inorganic P solubilization; (c) P uptake and transport systems; and (d) P starvation response regulation. This selection formed a new gene set specifically related to soil P cycling. The selected P cycling genes underwent function and taxonomy annotation against the KEGG and NCBI NR databases using Diamond. To determine the relative contribution of specific microbial taxa to soil P cycling, we calculated the abundance of the taxon of interest relative to the total abundance of all taxa involved in soil P cycling.

### 2.7. Statistical Analysis

Data organization was performed using Excel 2021. To investigate significant differences in the relative abundance and Shannon–Wiener index of genes related to soil P cycling among different plants, one-way analysis of variation (ANOVA) and least significant difference (LSD) multiple comparisons ($p < 0.05$) were conducted using SPSS 21.0 statistical software package (IBM, Armonk, NY, USA). To compare the difference between plants with high and low PPEIs in the relative abundance of functional genes, an independent *T*-test was conducted using the SPSS 21.0 statistical software package. The Pearson correlation analysis was employed to examine the correlation between the plant nutrients, plant P utilization efficiency index (PPEI), and rhizosphere soil properties of the studied plants. To investigate the response of the PPEI to the relative abundances of different categories of functional genes related to soil P cycling, linear regression analysis was conducted using the SPSS 21.0 statistical software package. For comparing the composition of P cycling-related microorganisms in different plant rhizospheres, taxonomy annotation results at the phylum level were used. A random forest (RF) model was used to analyze the microbial species that directly affected the PPEI based on the "randomForest" software package in R version 4.2.3. Structural equation modeling (SEM) was constructed using SPSS Amos version 26.0 (IBM, Armonk, NY, USA) to investigate the direct effects of the relative abundance and diversity of P cycling genes on the PPEI. The suitable fitting of a constructed structural equation model was achieved through a maximum likelihood evaluation with a non-significant Chi-square test (Chi-square/df < 3 and $p$-value > 0.05 for the model) and a comparative ft index (CFI > 0.95).

## 3. Results

### 3.1. The Relationships between the Plant P Utilization Efficiency Index (PPEI) and Soil Properties

Based on the results presented in Table S1, we selected three plants with a high PPEI (*ND*, *MB*, and *PF*) and three plants with a low PPEI (*TS*, *AK*, and *MR*) for subsequent analysis. The plant N and P contents, N:P ratio, and plant P utilization efficiency (PPEI) of the six studied plants are presented in Table 1. The average N content of plants with high a PPEI is 2.10%, which is lower than that of plants with a low PPEI (2.69%). However, plants with a high PPEI exhibit a significantly higher average P content (0.15%) compared to plants with a low PPEI (0.09%). The plant N:P ratio exhibited a completely contradictory trend compared to that of the PPEI (high, 15.31; low, 33.00).

**Table 1.** The PPEI, N and P contents, and N:P ratio of the studied plants.

| P Utilization Level | Plant Species | PPEI | Plant Nutrients (%) | | N:P |
|---|---|---|---|---|---|
| | | | N | P | |
| High | *ND* | 0.076 ± 0.026 | 3.70 ± 0.04 | 0.28 ± 0.01 | 13.22 ± 0.44 |
| | *MB* | 0.063 ± 0.003 | 1.26 ± 0.06 | 0.08 ± 0.01 | 15.79 ± 0.73 |
| | *PF* | 0.059 ± 0.003 | 1.35 ± 0.08 | 0.08 ± 0.01 | 16.91 ± 0.79 |
| | Average | 0.066 ± 0.008 | 2.10 ± 1.20 | 0.15 ± 0.10 | 15.31 ± 1.74 |
| Low | *TS* | 0.036 ± 0.002 | 2.74 ± 0.09 | 0.10 ± 0.004 | 27.43 ± 1.46 |
| | *AK* | 0.032 ± 0.002 | 3.72 ± 0.08 | 0.12 ± 0.01 | 31.12 ± 2.09 |
| | *MR* | 0.025 ± 0.002 | 1.61 ± 0.08 | 0.04 ± 0.004 | 40.46 ± 3.23 |
| | Average | 0.031 ± 0.005 | 2.69 ± 0.92 | 0.09 ± 0.04 | 33.00 ± 6.17 |

*ND*, *Nandina domestica* Thunb.; *MB*, *Mahonia bodinieri* Gagnep.; *PF*, *Pyracantha fortuneana* (Maxim.) Li; *TS*, *Tirpitzia sinensis* (Hemsl.) Hallier f.; *AK*, *Albizia kalkora* (Roxb.) Prain; *MR*, *Morella rubra* Lour.; PPEI, plant P utilization efficiency index. Data are expressed as means ± SD based on one-way ANOVA.

The PPEI exhibited a significant and positive correlation with plant P content ($R = 0.642$, $n = 18$, $p < 0.01$) while demonstrating a negative correlation with the plant N:P ratio ($R = -0.960$, $n = 18$, $p < 0.01$) (Table 2). Likewise, the plant P content displayed a significant positive correlation with plant N content ($R = 0.728$, $n = 18$, $p < 0.01$), but exhibited a negative correlation with the plant N:P ratio ($R = -0.556$, $n = 18$, $p < 0.05$). The

soil total N, total P, AN, and AP were significantly and positively correlated with each other ($R = 0.474$–$0.968$, $n = 18$, $p < 0.05$ or $p < 0.01$).

**Table 2.** Pearson correlation coefficients between plants and soils.

| Indexes | | Plant | | | | Soil | | | |
|---|---|---|---|---|---|---|---|---|---|
| | | **N** | **P** | **N:P** | **PPEI** | **Total N** | **Total P** | **AN** | **AP** |
| Plant | N | 1 | | | | | | | |
| | P | 0.728 ** | 1 | | | | | | |
| | N:P | 0.003 | −0.556 * | 1 | | | | | |
| | PPEI | 0.004 | 0.642 ** | −0.960 ** | 1 | | | | |
| Soil | Total N | 0.151 | 0.193 | 0.064 | −0.053 | 1 | | | |
| | Total P | 0.132 | −0.153 | 0.329 | −0.407 | 0.777 ** | 1 | | |
| | AN | 0.030 | 0.207 | −0.095 | 0.103 | 0.968 ** | 0.661 ** | 1 | |
| | AP | 0.316 | 0.527 * | −0.204 | 0.251 | 0.897 ** | 0.474 * | 0.926 ** | 1 |

PPEI, plant P utilization efficiency index; AN, available nitrogen; AP, available phosphorus. * and ** indicate significant correlations at $p < 0.05$ and $p < 0.01$, respectively.

### 3.2. The Response of PPEI to Functional Genes Involved in Soil P Cycling

#### 3.2.1. The Relative Abundance and Diversity of Genes Involved in Soil P Cycling

The relative abundance of functional genes involved in soil P cycling followed the sequence of *ND* (0.15%) ≥ *PF* (0.14%) = *MB* (0.14%) = *AK* (0.14%) = *MR* (0.13%) ≥ *TS* (0.12%) (Figure 1A). Regarding these genes, the average abundance in plants with a high PPEI is 0.15%, which is significantly higher than that in plants with a low PPEI (0.13%) ($p < 0.01$). The Shannon index of these functional genes ranged from 3.91 to 3.95, and no significant differences were observed either between different plants or between the two groups of plants with high and low PPEIs (Figure 1B).

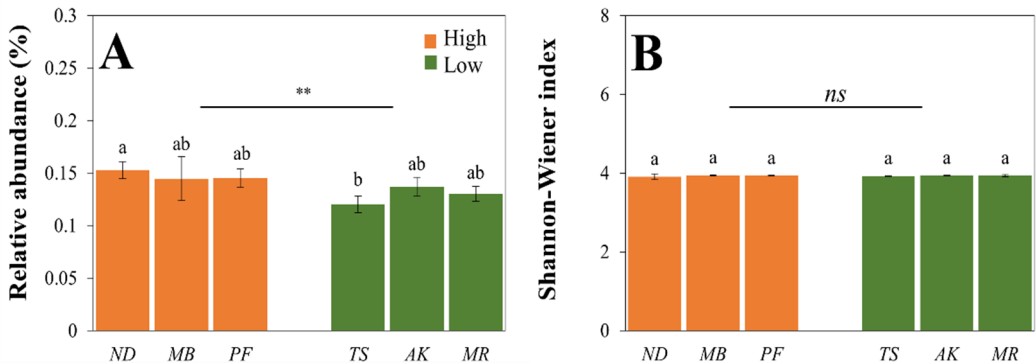

**Figure 1.** The relative abundance (**A**) and diversity (**B**) of microbial genes involved in soil P cycling. *ND*, *Nandina domestica* Thunb.; *MB*, *Mahonia bodinieri* Gagnep.; *PF*, *Pyracantha fortuneana* (Maxim.) Li; *TS*, *Tirpitzia sinensis* (Hemsl.) Hallier f.; *AK*, *Albizia kalkora* (Roxb.) Prain; *MR*, *Morella rubra* Lour. High, plants with a high P utilization efficiency index; Low, plants with a low P utilization efficiency index. Data are expressed as means ± SD; different small letters indicate a significant difference ($p < 0.05$) between the six studied plants based on a one-way ANOVA followed by an LSD test; ** and *ns* indicate significant ($p < 0.01$) and nonsignificant differences between two groups of plants with a high and low PPEI based on an independent T-test, respectively.

#### 3.2.2. The Response of the PPEI to the Relative Abundance and Diversity of Genes Involved in Soil P Cycling

The PPEI exhibits a positive linear relationship with the relative abundance of functional genes involved in soil P cycling ($R^2 = 0.378$, $p < 0.01$), indicating that microbial P cycling gene abundance significantly enhances PPEI (Figure 2A). However, there is no noticeable response of PPEI to the diversity of functional genes (Figure 2B).

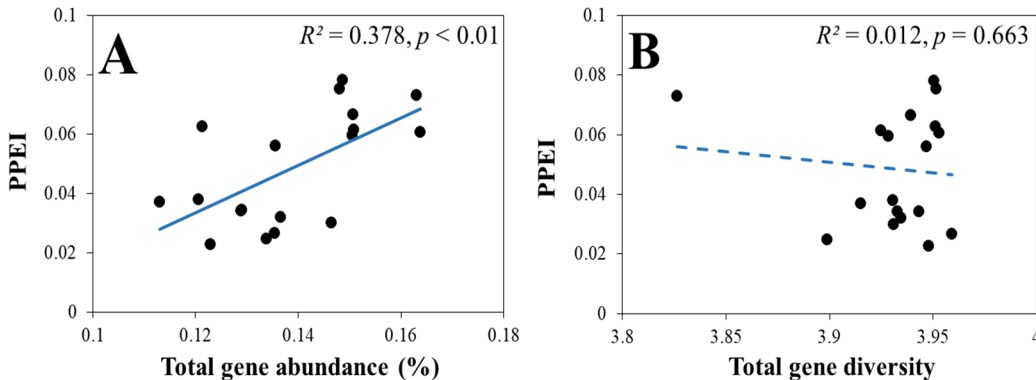

**Figure 2.** Relationships between the PPEI and the relative abundance (**A**) and the diversity (**B**) of microbial genes involved in soil P cycling. PPEI, plant P utilization efficiency index. Black-filled circles are the values of the tested samples. Solid and dashed blue lines indicate significant and nonsignificant results of linear regression analysis, respectively.

### 3.2.3. The Composition of Genes Involved in Soil P Cycling

When the genes were categorized into four P cycling groups, those involved in soil organic P mineralization exhibited the highest relative abundances across all plant rhizosphere soils, following this sequence: *ND* (0.073%) $\geq$ *MB* (0.067%) $\geq$ *PF* (0.066%) $\geq$ *AK* (0.063%) $\geq$ *MR* (0.060%) $\geq$ *TS* (0.056%) (Figure 3). The ranges of the relative abundances of genes participating in soil inorganic P solubilization, the soil P uptake and transport system, and soil P starvation response regulation were 0.032%–0.038%, 0.025%–0.032%, and 0.008%–0.011%, respectively, and no significant differences were observed among the six studied plants with varying P use efficiencies. However, when these plants were divided into two groups, those with a high PPEI (1.01%–6.88%) had significantly higher average abundances in all four P cycling groups compared to plants with a low PPEI (0.86%–5.97%) (Table S3).

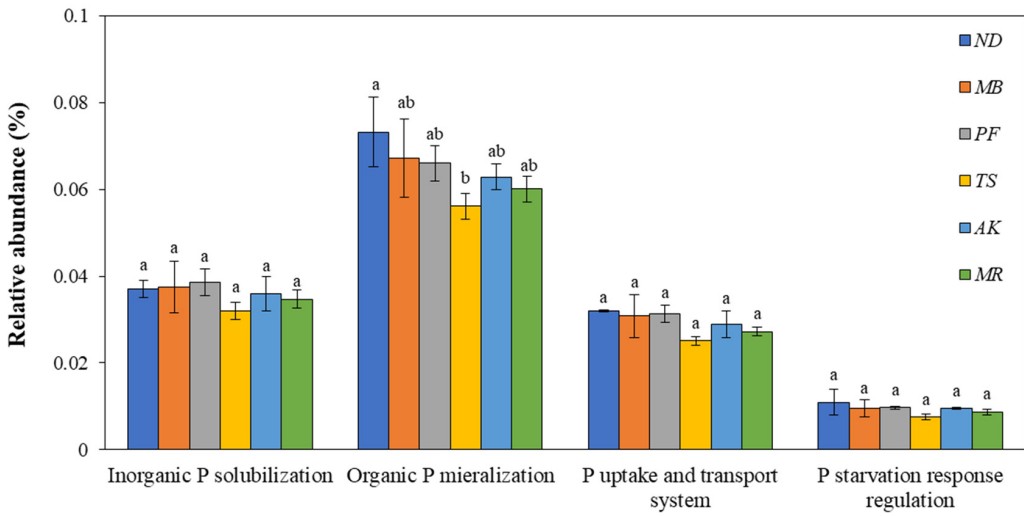

**Figure 3.** The relative abundances of the genes involved in four P cycling categories. *ND*, *Nandina domestica* Thunb.; *MB*, *Mahonia bodinieri* Gagnep.; *PF*, *Pyracantha fortuneana* (Maxim.) Li; *TS*, *Tirpitzia sinensis* (Hemsl.) Hallier f.; *AK*, *Albizia kalkora* (Roxb.) Prain; *MR*, *Morella rubra* Lour. Data are presented as means $\pm$ SD, and different small letters indicate a significant difference ($p < 0.05$) between the six studied plants based on a one-way ANOVA followed by an LSD test.

### 3.2.4. The Response of the PPEI to the Four Categories of Genes Involved in Soil P Cycling

Among the four different categories of genes participating in soil P cycling, the PPEI displayed a significant and positive response to the relative abundance of genes involved in soil organic P mineralization ($R^2 = 0.416$, $p < 0.01$) and the soil P uptake and transport system ($R^2 = 0.388$, $p < 0.01$) (Figure 4B,C). However, no positive response was found about the relative abundance of genes involved in soil inorganic P solubilization and P-starvation response regulation (Figure 4A,D).

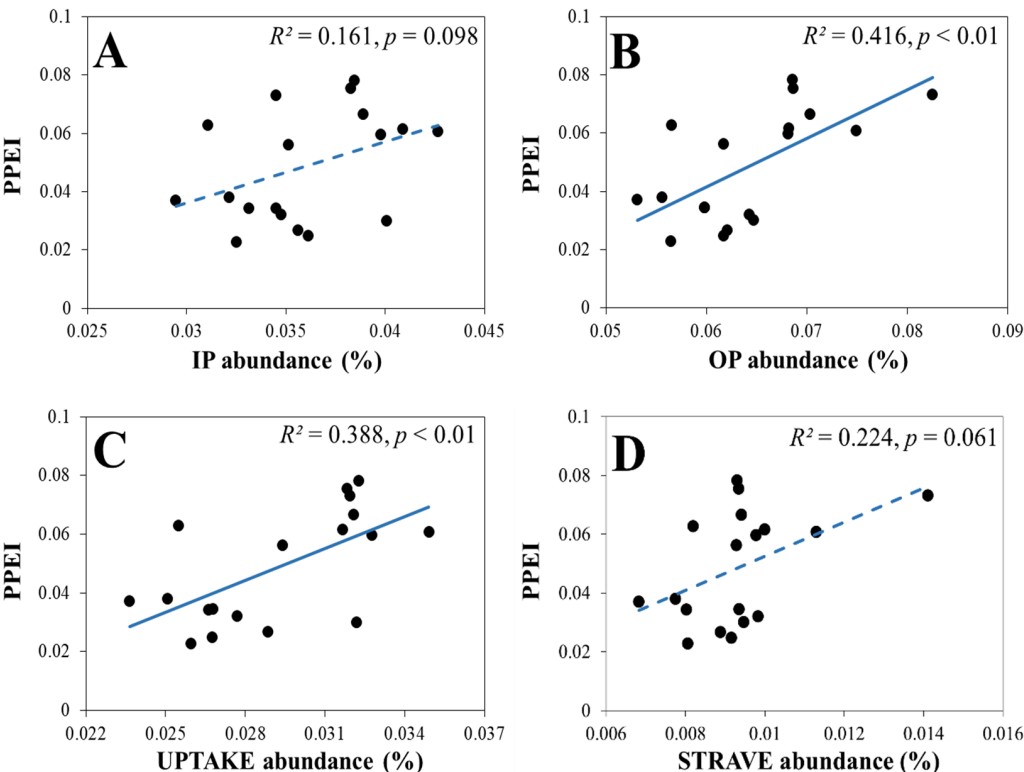

**Figure 4.** Relationships between the PPEI and the relative abundances of the genes involved in soil inorganic P solubilization (**A**), organic P mineralization (**B**), the P uptake and transport system (**C**), and P-starvation response regulation (**D**). PPEI, plant P utilization efficiency index; IP abundance, the relative abundance of genes involved in soil inorganic P solubilization; OP abundance, the relative abundance of genes involved in soil organic P mineralization; UPTAKE abundance, the relative abundance of genes involved in the soil P uptake and transport system; STRAVE abundance, the relative abundance of genes involved in soil P-starvation response regulation. Black-filled circles are the values of the tested samples. Solid and dashed blue lines indicate the significant and nonsignificant results of the linear regression analysis, respectively.

### 3.3. The Response of the PPEI to Genes Involved in Soil Organic P Mineralization

#### 3.3.1. The Composition of Genes Involved in Soil Organic P Mineralization

Previous results indicate that the PPEI significantly responds to the relative abundance of genes involved in soil organic P mineralization. In this study, a total of six sub-categories of functional genes were included in the microbial organic P mineralization (Figure 5). Among them, the relative abundances of the genes responsible for the synthesis of phosphomonoesterase were the only one that showed differences among plants and followed this sequence: *ND* (0.029%) ≥ *MB* (0.026%) ≥ *PF* (0.025%) ≥ *TS* (0.023%) = *MR* (0.023%) = *AK* (0.023%) (Figure 5). The relative abundances of the genes involved in the production of phosphodiesterase, organic pyrophosphatase, C-P lyase, phosphonatase, and phosphotriesterase ranged from 0.021% to 0.029%, 0.0059% to 0.0077%, 0.0033% to 0.0050%, 0.0014% to 0.0022%, and 0.00085% to 0.0010%, respectively, with no significant differences observed among the six studied plants (Figure 5). When these plants were divided into two groups,

those with a high PPEI had significantly higher average abundances of genes responsible for the synthesis of phosphomonoesterase (2.69%) and organic pyrophosphatase (0.72%), respectively, compared to plants with a low PPEI (phosphomonoesterase, 2.32%; organic pyrophosphatase, 0.61%) (Table S4).

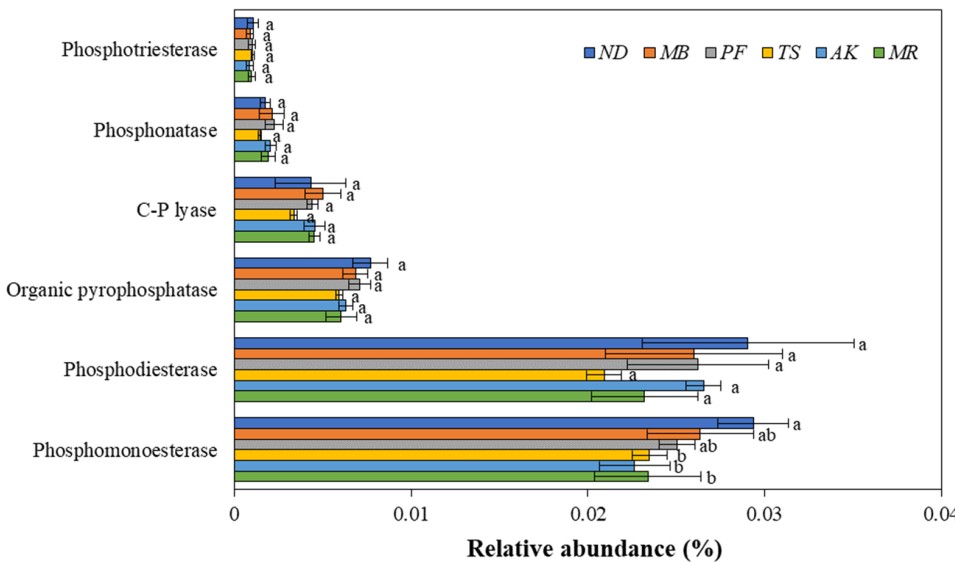

**Figure 5.** The relative abundances of the genes involved in soil organic P mineralization. *ND*, *Nandina domestica* Thunb.; *MB*, *Mahonia bodinieri* Gagnep.; *PF*, *Pyracantha fortuneana* (Maxim.) Li; *TS*, *Tirpitzia sinensis* (Hemsl.) Hallier f.; *AK*, *Albizia kalkora* (Roxb.) Prain; *MR*, *Morella rubra* Lour. Data are presented as means ± SD, and different small letters indicate a significant difference ($p < 0.05$) between the six studied plants based on a one-way ANOVA followed by an LSD test.

### 3.3.2. The Response of the PPEI to the Six Sub-Categories of Genes Involved in Soil Organic P Mineralization

Among the six sub-categories of genes participating in soil organic P mineralization, the PPEI exhibits a positive linear relationship with the relative abundances of genes involved in the production of phosphomonoesterase ($R^2 = 0.566$, $p < 0.01$) (Figure 6A) and organic pyrophosphatase ($R^2 = 0.418$, $p < 0.01$) (Figure 6F), while no positive relationship was found between the PPEI and the relative abundances of the other four categories of functional genes (phosphodiesterase, C-P lyase, phosphonatase, and phosphotriesterase) (Figure 6B–E).

### 3.3.3. The Response of the PPEI to the Genes Involved in the Production of Phosphomonoesterase and Organic Pyrophosphatase

In the present study, a total of 12 and 2 genes were identified to be involved in the production of phosphomonoesterase and organic pyrophosphatase, respectively (Table 3). Among the 12 genes responsible for the synthesis of phosphomonoesterase, the relative abundances of *G6PD*, *suhB*, and *phoD* exhibited positive linear relationships with the PPEI. In the case of the organic pyrophosphatase production, *ppx* showed a significant correlation with the PPEI (Table 3).

### 3.4. The Response of the PPEI to Genes Involved in the Soil P Uptake and Transport System
#### 3.4.1. The Composition of Genes Involved in the Soil P Uptake and Transport System

Previous results suggested that plant P utilization efficiency exhibits a positive linear relationship with the relative abundances of genes involved in the soil P uptake and transport system, which comprises four sub-categories of genes (Figure 7). Specifically, the relative abundances of the genes associated with phosphate (0.015%–0.020%), phosphonate (0.0063%–0.0080%), and IP phosphate (0.00004%–0.00015%) transport systems showed no

significant differences among the six studied plants. While the relative abundances of genes participating in the soil GP phosphate transport system followed the sequence of *PF* (0.0049%) ≥ *MB* (0.0043%) ≥ *AK* (0.0042%) ≥ *ND* (0.0041%) ≥ *MR* (0.0037%) ≥ *TS* (0.0032%) (Figure 7). When these plants were divided into two groups, those with a high PPEI had significantly higher average abundances of genes involved in the phosphate transport system (1.92%) and GP phosphate transport system (0.44%), compared to plants with a low PPEI (phosphate transport system, 1.64%; GP phosphate transport system, 0.37%) (Table S5).

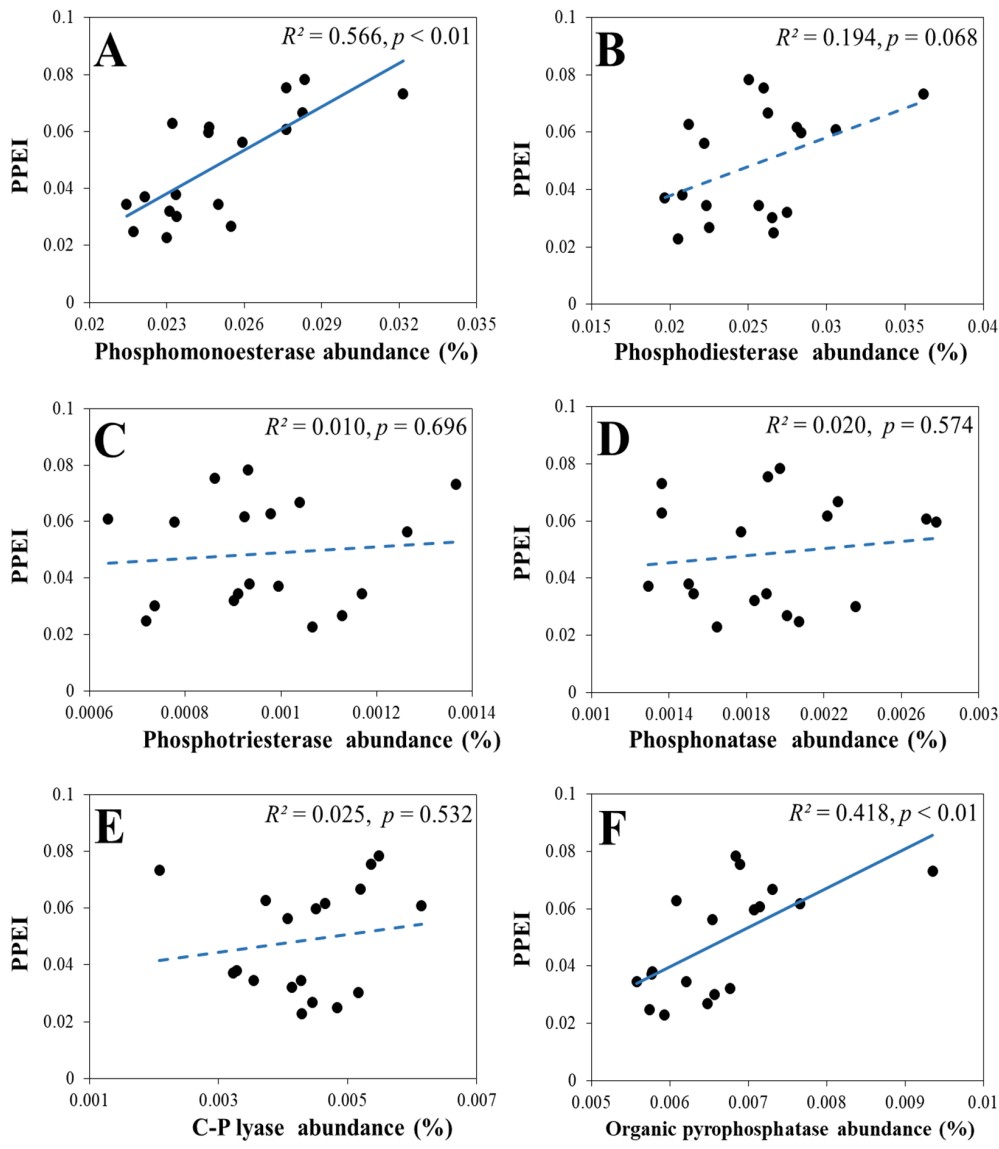

**Figure 6.** Relationships between the PPEI and the relative abundances of the genes involved in the production of phosphomonoesterase (**A**), phosphodiesterase (**B**), phosphotriesterase (**C**), phosphonatase (**D**), C-P lyase (**E**), and organic pyrophosphatase (**F**). PPEI, plant P utilization efficiency index. Black-filled circles are the values of the tested samples. Solid and dashed blue lines indicate the significant and nonsignificant results of the linear regression analysis, respectively.

**Table 3.** Relationships between the PPEI and the relative abundances of the genes involved in the production of phosphomonoesterase and organic pyrophosphatase.

| Gene Name | | $R^2$ | $p$-Value |
|---|---|---|---|
| | *G6PD* | 0.265 | <0.05 |
| | *gpsA* | 0.179 | 0.080 |
| | *gld* | 0.000 | 0.096 |
| | *phoA* | 0.006 | 0.752 |
| | *E3.1.3.8* | 0.158 | 0.102 |
| Phosphomonoesterase | *suhB* | 0.379 | <0.01 |
| | *appA* | 0.038 | 0.438 |
| | *pgpA* | 0.005 | 0.771 |
| | *phoD* | 0.453 | <0.01 |
| | *aphA* | 0.013 | 0.655 |
| | *phoX* | 0.080 | 0.254 |
| | *phoN* | 0.003 | 0.842 |
| Organic pyrophosphatase | *nudF* | 0.103 | 0.193 |
| | *ppx* | 0.450 | <0.01 |

$p$-value < 0.05 or <0.01 indicates a significant result of the linear regression analysis between the PPEI and the relative abundances of genes; otherwise, it was not considered significant. PPEI, plant P utilization efficiency index.

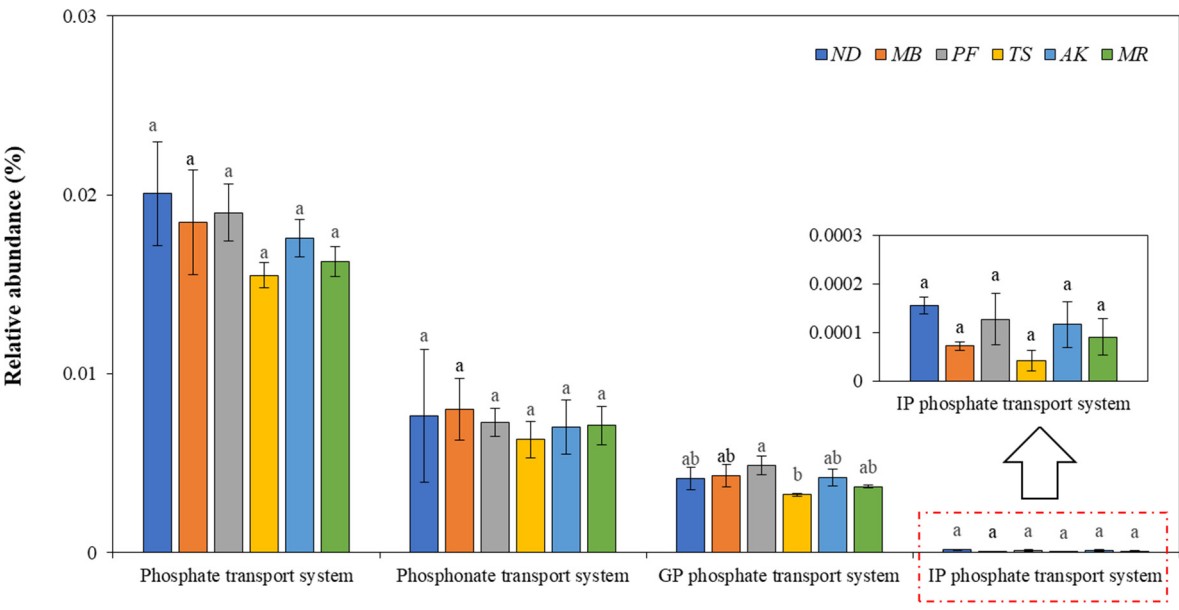

**Figure 7.** The relative abundances of the genes involved in the soil P uptake and transport system. *ND*, *Nandina domestica* Thunb.; *MB*, *Mahonia bodinieri* Gagnep.; *PF*, *Pyracantha fortuneana* (Maxim.) Li; *TS*, *Tirpitzia sinensis* (Hemsl.) Hallier f.; *AK*, *Albizia kalkora* (Roxb.) Prain; *MR*, *Morella rubra* Lour. GP, glycerol phosphate; IP, inositol phosphate. Data are presented as means ± SD, and different small letters indicate a significant difference ($p < 0.05$) between the six studied plants based on a one-way ANOVA followed by an LSD test.

### 3.4.2. The Response of the PPEI to the Four Sub-Categories of Genes Involved in the Soil P Uptake and Transport System

Among the four sub-categories of genes that participate in the soil P uptake and transport system, only the genes related to the phosphate transport system showed a positive linear relationship with the PPEI ($R^2 = 0.365$, $p < 0.01$) (Figure 8A). And no positive relationships were found between the PPEI and the relative abundances of the other three categories of functional genes (Figure 8B–D).

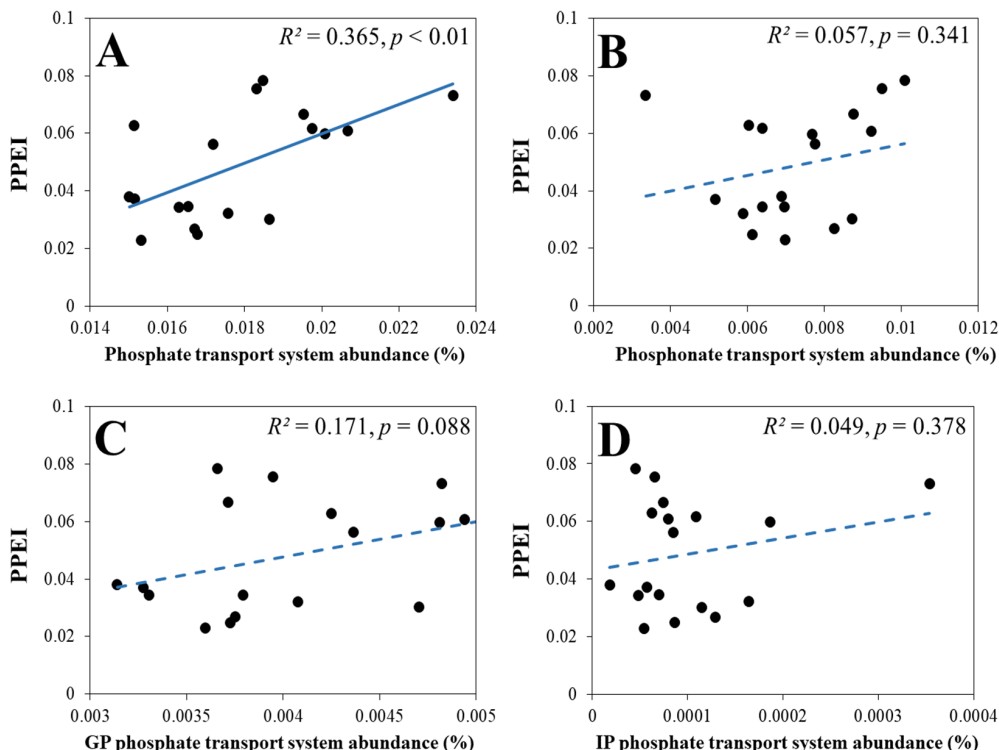

**Figure 8.** Relationships between the PPEI and the relative abundances of the genes involved in the GP phosphate (**A**), IP phosphate (**B**), phosphate (**C**), and phosphonate (**D**) transport systems. PPEI, plant P utilization efficiency index. Black-filled circles are the values of the tested samples. Solid and dashed blue lines indicate significant and nonsignificant results of the linear regression analysis, respectively.

### 3.4.3. The Response of the PPEI to the Genes Involved in the Soil Phosphate Transport System

A total of five genes were identified to be involved in the soil phosphate transport system (Table 4). Except for the *pit* genes, the relative abundances of *pst* (*pstS*, *pstA*, *pstB*, and *pstC*) were all positively correlated with the PPEI. When these plants were divided into two groups, those with high a PPEI exhibited significantly higher average abundances of *pstSABC* genes (0.017%) compared to plants with a low PPEI (0.015%), but no significant difference was observed in the relative abundance of the *pit* gene (0.0018% and 0.0016%) (Table S6).

**Table 4.** Relationships between the PPEI and the relative abundance of genes involved in the soil phosphate transport system.

|  | Gene Name | $R^2$ | *p*-Value |
|---|---|---|---|
| Phosphate transport system | *pstS* | 0.232 | <0.05 |
|  | *pstA* | 0.354 | <0.01 |
|  | *pstB* | 0.285 | <0.05 |
|  | *pstC* | 0.305 | <0.05 |
|  | *pit* | 0.025 | 0.527 |

*p*-value < 0.05 or <0.01 indicates the significant results of the linear regression analysis between the PPEI and the relative abundances of genes; otherwise, it was not considered significant. PPEI, plant P utilization efficiency index.

### 3.5. Taxonomic Composition and Contributions of Genes Involved in Soil P Cycling

In this study, 84 microbial phyla were identified to be involved in soil P cycling. For the six studied plants with different PPEIs, the top 10 predominant microbial phyla were consistent (Figure 9). Among them, the relative contribution of Proteobacteria ranked first in all

studied plant rhizosphere soils, ranging from 45.16% to 60.02%, followed by Actinobacteria (9.45%–25.23%), Acidobacteria (5.90%–9.85%), Candidatus_Rokubacteria (6.87%–7.68%), and Verrucomicrobia (3.83%–5.52%). The relative contributions of unclassified_d_Bacteria, Chloroflexi, Planctomycetes, and Cyanobacteria to P-cycling microbes in all studied soils were below 5% (Figure 9). The top 10 predominant microbial species involved in soil P cycling are presented in Table S7. Notably, six species belong to Proteobacteria, indicated again the substantial contribution of this microbial phylum to P cycling in the rhizosphere of karst plants. Further, an RF analysis revealed that *Betaproteobacteria_bacterium* and *Candidatus_Rokubacteria_bacterium* make significant contributions to the PPEI in the studied karst forest (Figure S1).

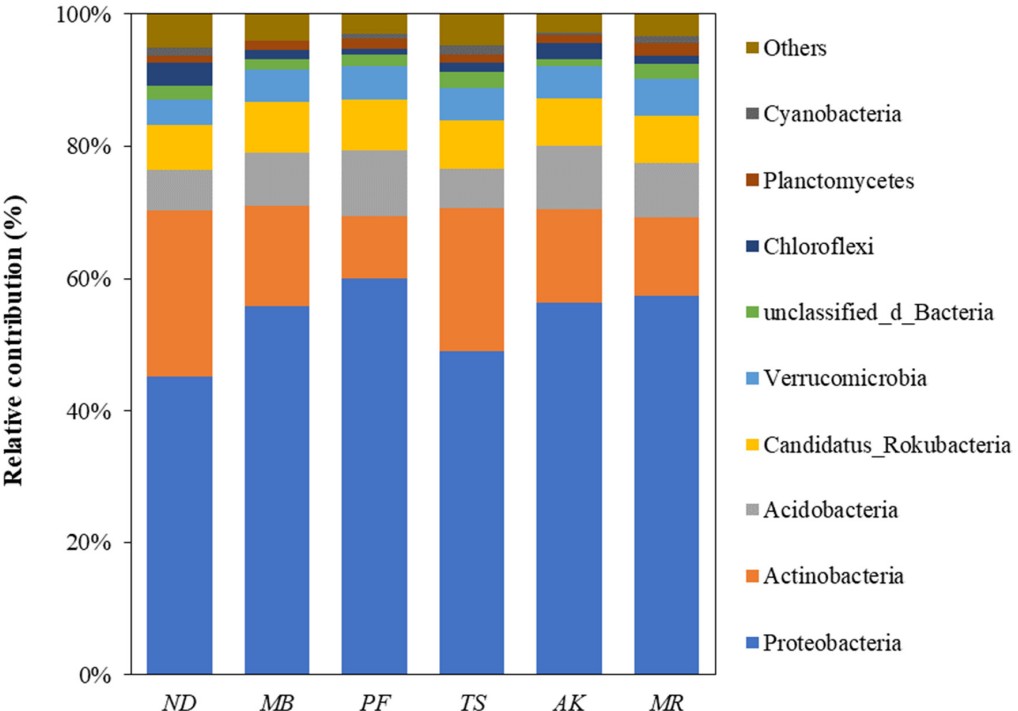

**Figure 9.** Contributions of soil microbial phyla to the genes involved in soil P cycling in different plant rhizosphere soils. *ND*, *Nandina domestica* Thunb.; *MB*, *Mahonia bodinieri* Gagnep.; *PF*, *Pyracantha fortuneana* (Maxim.) Li; *TS*, *Tirpitzia sinensis* (Hemsl.) Hallier f.; *AK*, *Albizia kalkora* (Roxb.) Prain; *MR*, *Morella rubra* Lour.

*3.6. The Direct Driving Factors of the PPEI*

A structural equation model (Figure 10) was constructed based on the results from the six studied plants and with a high goodness of fit. The structural equation model revealed that the relative abundances of genes involved in soil P cycling significantly and positively influenced the plant P utilization efficiency index (PPEI), but for plant N and P contents, the influences were not significant. In addition, the diversity of genes involved in soil P cycling showed no significant impact on either the PPEI, plant N content, or plant P content.

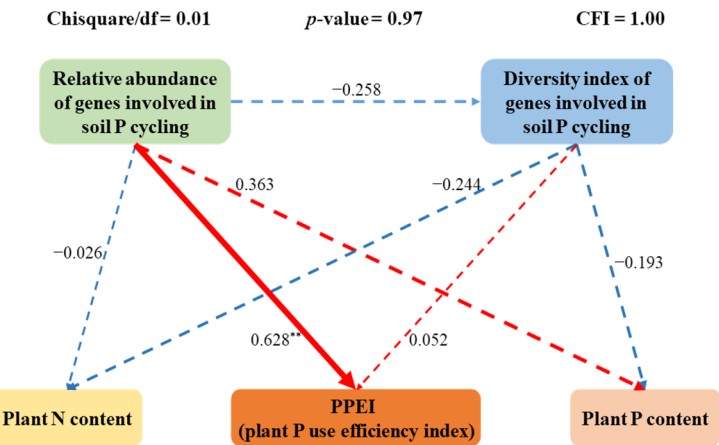

**Figure 10.** Direct and indirect effects of the relative abundances and the diversity index of genes involved in soil P cycling on the plant N and P contents and plant P utilization efficiency index. Rectangles represent observed variables. The arrows indicate the direction of the effect. The blue arrows represent positive effects and the red arrows represent negative effects. The numbers adjacent to the arrows indicate the effect size of the relationships. The width of an arrow is proportional to the strength of the path coefficient. Continuous and dashed arrows indicate significant ($p < 0.05$ or $p < 0.01$) and nonsignificant effects, respectively. The value of Chi-square/df < 3, $p$-value > 0.05, and the CFI (Comparative Fit Index) > 0.95 are considered to indicate a good goodness of fit. The asterisk after the number indicates the significance size (**, $p < 0.01$).

## 4. Discussion

### 4.1. Nutrient Contents of Karst Plants

Plant nutrient concentrations and stoichiometry have been extensively employed to assess nutrient utilization efficiency and identify the limiting factors for plant growth and terrestrial net primary production [56,66,67]. For instance, C consistently constitutes approximately 50% of plant dry biomass [68,69]. In contrast, N and P, both individually and interactively, serve as the primary limiting elements in terrestrial ecosystems, playing pivotal roles in various physiological and metabolic activities crucial for plant growth [70]. Within the study area, the mean leaf C and N concentrations in the karst plants were 464.6 mg/g and 19.1 mg/g, respectively (Table S1). These values are slightly higher than those found in global flora (C, 461.6 mg/g; N, 18.3 mg/g) and Chinese flora (C, 436.8 mg/g; N, 18.6 mg/g) [71–74]. However, the leaf P concentration in karst plants ranges from 0.4 mg/g to 2.8 mg/g, with a mean value of 0.9 mg/g, which is significantly lower than the Chinese and global averages (1.21 mg/g and 1.42 mg/g) [72,73]. In general, nutrient concentrations in plants and soil exhibit close associations across various ecosystem scales [75,76]. Therefore, in the present study, the observed coupling relationships between plant leaf P and soil-available P (Table 2, 0.527 *, $p < 0.05$) suggest that the deficient soil-available P may contribute to the low leaf P levels in karst plants. Furthermore, it is widely acknowledged that P limitation is prevalent in tropical/subtropical regions due to heightened weathering and the strong combination of available P and base cations [77]. In particular, in karst soils, available P reacts with calcium to form insoluble precipitates, resulting in lower levels of soil-available P compared to other ecosystems (Table S2, 9.12 mg/kg) [78]. In addition to these factors, climatic variations and soil properties significantly influence the P limitation in karst ecosystems, encompassing precipitation, seasonal climate patterns, soil pH, and texture. For instance, in karst ecosystems, rainwater is often slightly acidic due to dissolved carbon dioxide [79]. This acidity can interact with soluble rocks, leading to the leaching of minerals and ultimately influencing soil nutrient availability. Seasonal climate variations, including temperature and precipitation fluctuations, can impact microbial activity, organic matter decomposition rates, and subsequently influence plant nutrient uptake and limitation [80]. Phosphorus availability typically decreases in alkaline soils. Karst

soils are often alkaline due to the presence of limestone [42]. Therefore, in such alkaline environments, P tends to become less available, leading to plant-available P limitation. As a result, the leaf P concentration in karst plants is significantly lower than in other ecosystems, indicating a universal P limitation for karst plants [81–83]. Conversely, the higher leaf N concentration in karst plants may be attributed to the saturated soil-available N levels (Table S2, 261.1 mg/kg). This is also supported by a prior study that suggested that karst ecosystems may have reached N saturation due to their high soil N content, which ranks among the highest globally [43]. The elevated organic C concentration in karst soil (Table S2, 45.3 mg/kg) is likely attributed to the high leaf C content of karst plants [42–44]. It was noticed that a positive correlation exists between the leaf N and P concentrations (Table 2, 0.728 **, $p < 0.01$), which was one of the main appearances for the nutrient balance in the bodies of terrestrial higher plants and could be explained by the joint involvement of these two elements in the establishment of basic structures and the function realization in cells [84,85].

### 4.2. The Growth of Karst Plants Was Primarily Limited by P

Compared to plant nutrient concentrations, examining the stoichiometric ratios between nutrients provides a more accurate indication of nutrient limitation, cycling, and how plants respond to climate change and environmental conditions [86,87]. In analyzing the N and P contents and their corresponding N:P ratio, the limiting elements of organism growth, development, and reproduction can be effectively identified [88]. For example, the green leaf N:P ratio serves as a strong indicator of the relative nutrient availability to plants and has been extensively studied at regional and global levels in terrestrial ecosystems [56,57]. Generally, a leaf N:P < 14, between 14–16, and >16 indicate plant N limitation, N and P co-limitation, and P limitation, respectively [89]. In this study, as anticipated, the high leaf N and low P levels resulted in an average leaf N:P ratio of 22.73 (Table S1), significantly higher than 16, and notably different from the results observed in China's flora (14.4) and global flora (11.0 or 11.8) [71–73]. These findings indicate that the growth of karst plants was primarily limited by P rather than N, which is consistent with numerous previous studies [90,91]. Furthermore, the N:P ratio in leaves was often negatively correlated with the growth rate and biomass production of the plants [92]. A high leaf N:P ratio in karst plants also indicates the harsh karst environment for plant growth. Specifically, the leaf N:P ratio ranged from 13.21 in *Nandina domestica* Thunb. to 40.25 in *Morella rubra* Lour. among the 20 studied karst plants (Table S1), highlighting substantial variations in P limitation across plant species.

Furthermore, leaf N:P ratio in karst plants was significantly and negatively correlated with plant P content (−0.556 **, $p < 0.01$), but not with plant N content (Table 2). These results are consistent with the observations for grassland and woody species [93–95] and indicate again that the shortage of available P in karst soils significantly restricts plant growth. Meanwhile, the variation in soil N concentration had no impact on the nutrient limitation for karst plants, suggesting, to some extent, that N was in a state of relative stability and sufficiency. Thus, karst plants are generally constrained by P, but the degree of the P limitation varies depending on the plant species.

### 4.3. The Relative Abundances of Rhizosphere Functional Genes Involved in P Cycling Varied across Different Plants

Plants depend on soil nutrients to complete each stage of growth and development. In turn, root exudates released during plant growth enhance the soil environment and the availability of nutrients [96,97]. These crucial plant–soil interactions predominantly occur in the rhizosphere, within a millimeter from the root surface [98]. This leads to significant disparities in rhizosphere soil properties compared to the surrounding bulk soil, particularly across different plant species. Therefore, in this study, the variations in rhizosphere soil properties among the six studied plants (Table S2) could be primarily attributed to both the plant species themselves and their interactions with rhizosphere soils.

The availability of soil nutrients directly influences the growth and reproduction of microorganisms, thereby shaping soil microbial communities. Additionally, plant root exudates can act as available nutrients and chemical attractants for microorganisms, facilitating the colonization and reproduction of the specific microorganisms in the rhizosphere and on the root surface enriched with these exudates, thereby changing the soil microbial structure and ecological function [99,100]. Therefore, the structures of soil microbial communities can be influenced by both edaphic properties and plant species [29,101–103]. In this study, the relative abundances of rhizosphere functional genes involved in soil P cycling varied among different plants growing in karst environments (Figure 1A), probably attributed to the variations in vegetation types and soil nutrient contents (Table S2).

However, it was noticed that no significant difference was observed in the diversity of rhizosphere functional genes involved in soil P cycling among different plants (Figure 1B), indicating a similar composition of microbial species engaged in P cycling in the rhizospheres of different karst plants. Compared to rhizosphere soil, bulk soil is generally regarded as a "seed bank" since the root microbial community is primarily derived from the bulk soil microbes through horizontal transfer [104,105]. Therefore, the rhizosphere P cycling microorganisms in the different studied plants all originate from the "seed bank", and no variation was observed in their genetic diversity, suggesting that different karst plants have the capacity to attract nearly every species of microbes involved in soil P cycling from the "seed bank", but different plants may exhibit varying degrees of attractiveness to these functional microbes.

### 4.4. Plant Rhizosphere P Cycling Microbial Community Has a Significant Positive Effect on the Plant P Utilization Efficiency (PPEI)

The PPEI is primarily determined by the plant genotype, but it can also be influenced by environmental conditions, particularly the rhizosphere microbiome [106]. The interactions between plants and the rhizosphere microbiome are intricate. On one hand, plants primarily regulate their root microbiota through root exudates and the plant immune system; on the other hand, the root microbiota expands the habitable range and metabolic capacity of plants and participates in diverse processes, including plant development, nutrient absorption, and stress responses [107–111]. Thus, a plant's phenotype is the combined outcome of its genotype and its microbiome [112]. In this study, *ND*, *MB*, and *PF* plants exhibited higher abundances of rhizosphere functional genes involved in soil P cycling when compared to *TS*, *AK*, and *MR* plants (Figure 1A), suggesting that plants with a higher PPEI have a greater potential within their rhizosphere microbiome for transforming soil P compounds. A further correlation analysis revealed a significant influence of P cycling genes on plant P utilization efficiency (Figure 2A). The structural equation model also indicated that the driving path from genetic abundance to the PPEI was significant (Figure 10). These results emphasize that both the plant genotype and its associated rhizosphere functional microbiome co-contribute to the P utilization efficiency of plants in karst environments. Given the absence of discernible differences in genetic diversity among various plant species (Figure 1B), it is reasonable to conclude that the diversity of functional genes involved in soil P cycling does not notably affect the PPEI of karst plants (Figure 10). Additionally, it is worth noting that despite substantial differences in leaf P and N contents among the six studied plants (Table 1), there is no evidence to suggest that rhizosphere P cycling microorganisms directly influence these outcomes (Figure 10). Therefore, rhizosphere microbes do not directly affect the absorption or absolute contents of plant N and P. Instead, they work to maintain a balance between N and P nutrients, alleviating P limitation and enhancing the P utilization efficiency of karst plants [106,113].

Under specific stress conditions, the host plant has the capacity to cultivate a distinct root-associated microbial community, which, in turn, may enhance the plant's resilience to that particular stressor [106]. In the studied karst region, the prevalent scarcity of available soil P has imposed a widespread P limitation on plant growth, thereby leading to the development of specialized rhizosphere microbiomes adept at coping with the stress of

low soil-accessible P levels. Nevertheless, it is important to note that the composition and ecological activity of these rhizosphere microbiomes engaged in P cycling are primarily influenced by the plants themselves, achieved through control over the type and quantity of root exudates that they release [99,100]. Therefore, in karst environments, aside from the inherent genetic characteristics, rhizosphere P cycling microorganisms can also play a significant role in enhancing plant P utilization efficiency. This dual support from both genotype and rhizosphere microorganisms collaboratively contributes to the adaptability of plants with a high PPEI in the face of low soil-available P stress under karst environments.

*4.5. The Specific P Cycling Genes Involved in the Alleviation of the P Limitation for Karst Plants*

4.5.1. Genes Involved in Soil Organic P Mineralization

Among the four P cycling categories presented in Figure 3, genes related to soil organic P mineralization exhibited the highest relative abundance in the rhizosphere soils of all studied plants. This finding aligns with the results obtained in [114], where it was discovered that genes responsible for organic P mineralization dominate in karst soils during the natural restoration of degraded karst vegetation. This trend was also observed in grassland, cropland, and tropical forests [12,115]. Previous studies have demonstrated that organic P mineralization is the primary driver of soil turnover in P-depleted soils [12,16]. Therefore, in the studied karst environments, the deficiency of plant-accessible P in rhizosphere soils (Table S2) likely leads to organic P mineralization dominating the soil P cycling process [116–118]. This deficiency of soil-available P has significantly impacted ecological functions in the karst regions of southwest China [119]. Under natural environmental conditions without artificial intervention, the plant-accessible P in soils mainly results from microbial transformations. In this study, plants with a high PPEI displayed a significantly higher relative abundance of genes associated with soil organic P mineralization compared to plants with a low PPEI ($p < 0.01$, Table S3). Further correlational analysis revealed that the increase in these functional genes contributes to the enhancement of the PPEI (Figure 4B). These findings suggest that the augmented rhizosphere functional microorganisms engaged in soil organic P mineralization alleviate the P limitation and enhance plant P utilization efficiency in karst areas. This could be attributed to two possible reasons: firstly, the higher SOC content in the rhizosphere soils of plants with a high PPEI (Table S2) renders organic P the dominant P source; and secondly, plants with a high PPEI may attract a greater number of microorganisms involved in soil organic P mineralization by manipulating their root exudates [22].

Microorganisms participate in soil organic P mineralization through the production of various phosphatases, which could be classified into six main categories including phosphomonoesterase, phosphodiesterase, organic pyrophosphatase, C-P lyase, phosphonatase, and phosphotriesterase (Figure 5). These enzymes are responsible for the hydrolysis of different organic compounds containing P. For example, phosphomonoesterase is an enzyme type that catalyzes the hydrolysis of phosphomonoesters, which are chemical compounds containing a single phosphate group attached to a molecule and are the primary form of organic compounds in soil [120]. In this study, plants with a high PPEI (*ND*, *MB*, and *PF*) exhibited significantly higher abundances of microbial functional genes involved in phosphomonoesterase production in their rhizosphere soils compared to plants with a low PPEI (*TS*, *AK*, and *MR*) (Figure 5, Table S4). Further correlational analysis revealed a significant and positive relationship between the relative abundance of microbial genes involved in phosphomonoesterase production and the plant PPEI (Figure 6A). These findings indicate that *ND*, *MB*, and *PF* possess a higher potential for rhizosphere microbial hydrolysis of soil phosphomonoesters compared to *TS*, *AK*, and *MR*. Moreover, the microbial potential for soil organic P mineralization also contributes to plant P utilization efficiency, thereby mitigating P stress in karst environments. Specifically, among the 12 genes involved in phosphomonoesterase production in this study, the relative abundances of *G6PD*, *suhB*, and *phoD* showed positive correlations with associated PPEIs (Table 3). These genes guide the synthesis of glucose-6-phosphate 1-dehydrogenase (EC: 1.1.1.49), myo-

inositol-1(or 4)-monophosphatase (EC: 3.1.3.25), and alkaline phosphatase (EC: 3.1.3.1), respectively. Of these three phosphomonoesterases, alkaline phosphatases are predominantly derived from soil microbes and are non-specific enzymes that catalyze the hydrolysis of ester–phosphate bonds in various orthophosphate monoesters, thereby enhancing P bioavailability in soils [121–123]. Additionally, the synthesis-guiding gene *phoD* is the most widespread in soils and serves as a reference marker in studies on soil P cycling [124]. In this study, gene *nudF* and *ppx* were identified to be involved in the production of organic pyrophosphatase, and only the relative abundance of *ppx* showed a positive correlation with the PPEI (Table 3). The *ppx* gene guides the synthesis of exopolyphosphatase (EC: 3.6.1.11), an enzyme that catalyzes the hydrolysis of inorganic polyphosphates into shorter-chain phosphates, typically orthophosphates [125]. Therefore, *ND*, *MB*, and *PF* exhibit a higher relative abundance of *G6PD*, *suhB*, *phoD*, and *ppx* genes than *TS*, *AK*, and *MR* in their rhizosphere soils, indicating that plants with a high PPEI can harbor a greater number of microorganisms capable of producing organophosphatases in their rhizospheres, which, in turn, can alleviate the plants' P limitation under karst environments.

#### 4.5.2. Genes Involved in the Soil P Uptake and Transport System

In general, soil functional genes involved in the soil P uptake and transport system are mainly influenced by the environmental P supply [126]. In this study, the relative abundance of genes related to this P cycling category did not show significant differences among the six studied plants (Figure 3). This may be attributed to the fact that plants in karst environments commonly experience a deficiency in available soil P, which has been confirmed by numerous previous studies [116–118].

Among the four sub-groups involved in P uptake and transport, genes involved in phosphate transport exhibited the highest relative abundance and showed a positive correlation with the PPEI, followed by phosphonate, glycerol phosphate, and inositol phosphate transporters (Figures 7 and 8). This observation can be attributed to variations in the content of different available P components in karst soils (Table S2). Inorganic phosphate likely constitutes the primary form of available P in karst soils, such as those found in the Tibetan Plateau [127]. Consequently, a high microbial potential for phosphate transport can thereby alleviate P limitation stress and enhance plant P utilization efficiencies.

Specifically, the phosphate transport system comprises two types of genes: the high-affinity phosphate-specific transporters (*pstSABC*, including *pstS*, *pstA*, *pstB*, and *pstC*) and the low-affinity phosphate-specific transporters (*pit*). These systems enable microbes to assimilate inorganic phosphate under conditions of P-rich and P-low environments, respectively [12,18]. In this study, no significant differences were observed in the relative abundance of *pit* between the two groups of plants (Table 4). However, plants with a high PPEI exhibited a significantly higher relative abundance of *pstSABC* compared to plants with a low PPEI (Table S6). Furthermore, the relative abundances of *pstSABC* all presented positive linear relationships with the PPEI (Table 4). These results suggest that plants in karst environments generally contend with a deficiency in available P. Nevertheless, the microbial uptake and transport of P through high-affinity transporters can mitigate this situation. Therefore, soil microbial high-affinity phosphate-specific transporters (*pstSABC*) play a crucial role in enhancing the P utilization efficiency of karst plants.

#### 4.6. Microbial Taxa Involved in Soil P Cycling

A total of 84 microbial phyla were identified in this study, highlighting the wide distribution of microbes involved in soil P cycling. Furthermore, the top 10 dominant microbial phyla (i.e., Proteobacteria, Actinobacteria, and Acidobacteria) remained consistent across all studied rhizosphere soils (Figure 9). The same results were also observed in the restoration soils in karst areas [114], the loess hilly region [128], and the Mu Us desert [129]. These findings suggest the strong competitive abilities of these microbial taxa across diverse environmental conditions. However, the composition of these microbial phyla was greatly influenced by different plant species. For example, Proteobacteria constituted 60.02% of *PF*,

whereas it accounted for 45.16% of *ND*; Actinobacteria represented 25.23% of *ND*, but only 9.45% of *PF*. The variations in microbial composition among the six studied plants may be attributed to differences in soil properties and the variability in plant root exudates.

## 5. Conclusions

Plant P limitation is a prevalent phenomenon in karst regions, primarily attributed to server soil-available P deficiency. Within karst environments, plants exhibiting high P utilization efficiencies demonstrate an enhanced capacity to recruit microorganisms engaged in soil P cycling within their rhizosphere, thereby displaying greater potential for the transformation of soil P compounds. Consequently, these functional genes play a direct role in augmenting plant P utilization efficiencies, independent of P contents. Specifically, several genes involved in soil organic P mineralization (*G6PD*, *suhB*, *phoD*, and *ppx*) and the P uptake and transform system (*pstS*, *pstA*, *pstB*, and *pstC*) drive this process. Proteobacteria, Actinobacteria, and Acidobacteria, among others, emerge as principal contributors to these P cycling genes across all examined plant rhizosphere soils. In conclusion, rhizosphere microbial P cycling functions serve to alleviate P limitation in karst plants. Both the plant genotype and its associated rhizosphere functional microbiome synergistically contribute to the P utilization efficiency of plants in karst environments. This research demonstrates the efficacy of rhizosphere microbial P cycling functions in enhancing the P utilization efficiency of karst plants and the associated underlying mechanisms from genetic and microbial perspectives. It provides valuable insights for comprehending and conserving karst forest ecosystems by guiding the selection of plants for restoration. Moreover, it underscores the pivotal role of rhizosphere microbial communities in nutrient cycling, offering perspectives for enhanced phosphorus management through potential strategies such as microbial inoculants. Future strategies and policies aimed at the restoration of karst vegetation may give enough attention to the strong plant–microorganism interactions. Subsequent investigations should focus on discerning the pivotal roles played by specific functional microbes in mitigating plant P limitations and enhancing plant P utilization efficiencies within the karst ecosystem.

**Supplementary Materials:** The following supporting information can be downloaded at https://www.mdpi.com/article/10.3390/f15030453/s1, Table S1: Plant nutrient contents, N:P, and plant P utilization efficiency index (PPEI). Table S2: Rhizosphere soil properties of the 20 investigated plants. Table S3: The average relative abundance (%) of genes involved in the four P cycling groups for plants with high and low PPEI. Table S4: The average relative abundance (%) of genes involved in the six organic P mineralization groups for plants with high and low PPEI. Table S5: The average relative abundance (%) of genes involved in the four P uptake and transport system groups for plants with high and low PPEI. Table S6: The average relative abundance (%) of genes involved in the four P uptake and transport system groups for plants with high and low PPEI. Table S7: The relative abundance (%) of top 10 predominant microbial species involved in P cycling. Figure S1: Relative importance of the relative abundance of the top 10 microbial species to the PPEI. The dark blue color indicates a significant effect ($p < 0.05$) and the light blue indicates not significant. The higher the %IncMSE value, the more important the variable. PPEI, plant P utilization efficiency index.

**Author Contributions:** Methodology, D.C., L.Z., G.Z., Q.L., M.S. and Y.H.; Formal analysis, C.Z., S.W., Y.D., L.W., R.B., Z.F. and F.X.; Investigation, C.Z.; Writing—original draft, C.Z.; Writing—review & editing, C.Z., D.C., L.Z., G.Z., Q.L., M.S. and Y.H.; Visualization, L.Z., G.Z., Q.L., M.S. and Y.H.; Supervision, D.C.; Project administration, D.C.; Funding acquisition, D.C. All authors have read and agreed to the published version of the manuscript.

**Funding:** This research was supported by the Basic Research Program in Guizhou Province [(2022)036]; the Gui Da Te Gang He Zi Program [(2021)06]; the Cultivation Project of Guizhou University [(2020)04]; the National Natural Science Foundation of China (32360380; 32360278); and the Guizhou University Scientific Research Innovation Team Project [(2023)07].

**Data Availability Statement:** The data will be made available by the authors on request.

**Acknowledgments:** Figure support was provided by Figdraw.

**Conflicts of Interest:** The authors declare that they have no known competing financial interests or personal relationships that could have appeared to influence the work reported in this paper.

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
