# Peer review of "The Rhizosphere Functional Microbial Community: A Key Driver of Phosphorus Utilization Efficiency in Karst Forest Plants"

_forests, doi:10.3390/f15030453_

Round 1

Reviewer 1 Report

Comments and Suggestions for Authors

·       Remove keywords that duplicate information from the title of the paper.

·       What was the size of the research sample for each selected plant and rhizosphere soil samples ?

·       The result in table 1S should be represented as mean ± SD or mean ± SE

·       Is there any difference in interactions between the microbial community in the rhizosphere and other soil microorganisms in the karst forest?

·       What are the potential implications of this research to understand and protect karst forest ecosystems?

·       How can the insights from this research be used to develop sustainable nutrient management strategies for karst forest ecosystems?

·       Can the microbial community in the rhizosphere be manipulated to enhance the phosphorus utilization efficiency in karst forests plants?

·       What are the possible long-term effects of changes in the rhizosphere microbial community on phosphorous utilization and ecosystem dynamics in karst forest ecosystems?

Reviewer 2 Report

Comments and Suggestions for Authors

Comments to authors:

The authors have conducted a comprehensive study exploring nutrient limitations in karst regions, focusing on phosphorus (P) deficiency and its impact on plant growth. The manuscript presents a thorough analysis of nutrient contents, stoichiometric ratios, and microbial taxa involved in soil P cycling. The study sheds light on the interactions between plant-microorganisms and their collective contribution to phosphorus utilization efficiency in karst environments. The English language in the content is generally clear and well-structured. The authors have effectively communicated their scientific findings, and the manuscript is written in a formal and scholarly tone. The language is precise, and the use of technical terms is appropriate for the target audience. However, while the manuscript is comprehensive and well-structured, there are a few areas where improvements can be considered:

1.      The study notes unexpected findings, such as the high leaf N:P ratio in karst plants. A more in-depth discussion of these contrasting results and their potential implications would enrich the interpretation.

2.      Further clarification on the specific mechanisms through which microbial communities contribute to phosphorus utilization efficiency would enhance the applicability of the findings.

3.      Consider discussing and addressing potential confounding factors that could influence the observed nutrient limitations, such as climatic variations or soil properties not explicitly considered in the study.

Several critical queries emerge from the study that could be addressed to further strengthen the findings and contribute to the broader scientific understanding:

1.      How does the functional diversity of microbial communities in the rhizosphere contribute to phosphorus utilization efficiency, and are there specific microbial functions that play a more dominant role?

2.      Does the observed enhancement in P utilization efficiency through plant-microbe interactions have long-term effects on the overall ecosystem structure and functioning?

3.      Given the heterogeneity of karst landscapes, how does the spatial variability in soil properties influence the extent of phosphorus limitation across different plant species?

4.      To what extent do anthropogenic factors, such as land-use changes or human activities, influence the observed nutrient limitations in karst ecosystems?

5.      How applicable are the study findings to other ecosystems with similar geological features, and are there lessons that can be extrapolated to guide nutrient management in diverse environments?

Addressing these queries could contribute to a more comprehensive understanding of the mechanisms underlying nutrient limitations in karst ecosystems and the implications for plant-microbe interactions.

 I hereby recommend for acceptance after minor revision of the manuscript. 

Thanks and regards,

‘The reviewer.

Reviewer 3 Report

Comments and Suggestions for Authors

The Rhizosphere Functional Microbial Community: A Key Driver of Phosphorus Utilization Efficiency in Karst Forest Plants

The article by Chunjie Zhou et al. describes  the study of Rhizosphere Functional Microbial Community involved in phosphorus metabolism in  P-deficient ecosystem of  karst forest located in Guizhou Province, China. The authors analyzed phosphorus content  as well as phosphorus/nitrogen ratios in leaves of local woody plants, and sampled the soil for microbiome analysis. The authors used several species of local flora with high and low levels of effective phosphorus utilization (P utilization efficiency index (PPEI).

The introduction is clearly and logically written and reflects the state of the art in this field of research.

To elucidate the status of microbial communities involved in phosphorus metabolism the authors performed Soil DNA Extraction and Metagenomic Sequencing, created a DNA Library and sequenced samples using Illumina NovaSeq. 

ORFs within the contigs were identified, then translated into amino acid sequences using the NCBI translation table. The obtained reads were aligned to the non-redundant gene catalog, with a 95% identity cutoff. Gene abundance in each sample was assessed based on the alignment results. For annotation, representative sequences from the non-redundant gene catalog were compared against the NCBI NR database using BLAST.

Authors selected 119 functional bacterial genes with their corresponding KO numbers from the KEGG database. These genes were attributed to four functional  categories: a) organic P mineralization; b) inorganic P solubilization; c) P uptake and transport systems; and d) P starvation response regulation. This selection formed a new gene set specifically related to soil P cycling. To determine the relative contribution of specific microbial taxa to soil P cycling, Authors  calculated the abundance of the taxon of interest relative to the total abundance of all taxa involved in soil P cycling.

In analyzing the results, the authors compared plant N and P accumulation levels, N:P ratio, and plant P utilization efficiency (PPEI) and correlated these factors with Relative Abundance and Diversity of bacterial genes involved in soil P cycling. The analyses showed that plants were phosphorus deficient but not nitrogen limited, with  an average leaf N:P ratio, that is notably different from results observed in China's flora and global flora. The analyses also allowed the authors to identify key bacterial genes affecting phosphorus availability. The authors found that PPEI displayed a significant and positive response to the relative abundance of genes involved in soil organic P mineralization and soil P uptake and transport system, but not with genes involved in soil inorganic P solubilization and P starvation response regulation.

Regarding genes involved in phosphorus mineralization, the authors showed that phosphomonoesterase and organic pyrophosphatase gene synthesis showed differences among analyzed plants and identified functional genes responsible for the synthesis of these enzymes. The authors also identified genes of phosphate transport system that showed a positive linear relationship with PPEI.

In the next step of the work, the authors, after sequencing analysis, were able to perform a taxonomic analysis of the bacteria that are involved in phosphorus metabolism and metabolism and created the Structural equation model that allowed the analysis of the factors of PPEI. The PPEI is primarily determined by the plant genotype, but it can also be influenced by environmental conditions, particularly the rhizosphere microbiome.

The results and discussion are clearly and accessibly presented and aim oriented.

I would also like to note the good planning of the experiments, the logical formulation of the research objectives, and the use of well-tested and reproducible methods of soil and plant analyses in combination with molecular biology and bioinformatics methods to analyze the microbiome.   The microbiome data obtained by the authors will undoubtedly be in demand in further research on this topic. I think, that the potential use of a "bouquet" of several bacterial species to improve the phosphorus availability may open up new technological opportunities for agriculture and business.

The advantage of this article is that the authors address the very urgent problem of phosphorus deficiency. Phosphorus is an exhaustible resource for the Earth's biosphere, due to its intensive use in agriculture.  Phosphorus cannot be replenished like nitrogen by fixation from the air, and its resources:   mineral salts and cattle manure may be exhausted soon. Therefore, studies such as those of the authors to increase phosphorus availability through the use of some soil bacteria are very relevant.

The authors have done good experimental work and written a good paper with interesting and practically useful results.

Comments on the Quality of English Language

The article is written in good English, but it will in any case be read and corrected if necessary by a native-speaking editor

Author Response

Thanks for your comments!